# Infrared and Visible Image Fusion Techniques Based on Deep Learning: A Review

**Changqi Sun** [1]‍, **Cong Zhang** [1],*‍ **and Naixue Xiong** [2],*‍

[1]	School of Mathematics and Computer Science, Wuhan Polytechnic University, Wuhan 430023, China;
	cqisun@yeah.net
[2]	Department of Mathematics and Computer Science, Northeastern State University, Tahlequah,
	OK 74464, USA
*	Correspondence: hb_wh_zc@163.com (C.Z.); xiongnaixue@gmail.com (N.X.)

**Abstract:** Infrared and visible image fusion technologies make full use of different image features obtained by different sensors, retain complementary information of the source images during the fusion process, and use redundant information to improve the credibility of the fusion image. In recent years, many researchers have used deep learning methods (DL) to explore the field of image fusion and found that applying DL has improved the time-consuming efficiency of the model and the fusion effect. However, DL includes many branches, and there is currently no detailed investigation of deep learning methods in image fusion. In this work, this survey reports on the development of image fusion algorithms based on deep learning in recent years. Specifically, this paper first conducts a detailed investigation on the fusion method of infrared and visible images based on deep learning, compares the existing fusion algorithms qualitatively and quantitatively with the existing fusion quality indicators, and discusses various fusions. The main contribution, advantages, and disadvantages of the algorithm. Finally, the research status of infrared and visible image fusion is summarized, and future work has prospected. This research can help us realize many image fusion methods in recent years and lay the foundation for future research work.

**Keywords:** image fusion; visible image; infrared image; evaluation metric; generative adversarial network

---

## 1. Introduction

Under normal conditions, objects will radiate electromagnetic waves of different frequencies, which is called thermal radiation. It is difficult for people to see thermal radiation information with the naked eye [1]. It is necessary to use different sensors [2–10] to process the infrared image to obtain its thermal radiation information, which has good target detection ability [11]. Infrared images can avoid the influence of the external environment, such as sunlight, smoke, and other conditions [1,12]. However, infrared images have low contrast, complex background, and poor feature performance. Visible images are consistent with the human eye's visual characteristics and contain many edge features and detailed information [13]. The use of visible light sensors to obtain image spectral information is richer, scene details and textures are clear, and spatial resolution is high. However, due to the external environment's influence, such as night environment, camouflage, smoke hidden objects, background clutter, etc., the target may not be easily observed in the visible image. Therefore, infrared and visible light fusion technology combines the two's advantages and retains more infrared and visible feature information in the fusion result [14]. Due to the universality and complementarity of infrared images and visible images, the fusion technology of infrared and visible images has been applied to more fields and plays an increasingly important role in computer vision. Nowadays, the fusion method of infrared and visible images have been widely used in

target detection [15], target recognition [16], image enhancement [17], remote sensing detection [18], agricultural automation [19,20], medical imaging [21], industrial applications [22–24].

According to different image fusion processing domains, image fusion can be roughly divided into two categories: the spatial and transform domains. The focus of the fusion method is to extract relevant information from the source image and merge it [25]. Current fusion algorithms can be divided into seven categories, namely, multi-scale transform [26], sparse representation [27], neural network [28], subspace [29], saliency [30], hybrid models [31], and deep learning [32]. Each type of fusion method involves three key challenges, i.e., image transform, activity-level measurement, and fusion rule designing [33]. Image transformation includes different multiscale decomposition, various sparse representation methods, non-downsampling methods, and a combination of different transformations. The goal of activity level measurement is to obtain quantitative information to assign weights from different sources [12]. The fusion rules include the big rule and the weighted average rule, the essence of which plays the role of weight distribution [32]. With the rapid development of fusion algorithms in theory and application, selecting an appropriate feature extraction strategy is the key to image fusion. It is still challenging to design a suitable convolutional neural network and adjust the parameters based on deep learning image fusion. Especially in recent years, after generating a confrontation network for image fusion, although it brings a clearer fusion effect, it also needs to consider the inevitable gradient disappearance and gradient explosion of the generation confrontation training.

In the field of image fusion, a variety of different infrared and visible image fusion methods have been proposed in recent years. However, there are still some challenges in different infrared and visible image fusion applications. The commonly seen fusion method is to select the same salient features of the source image and integrate them into the fusion image to contain more detailed information. However, the infrared heat radiation information is mainly characterized by pixel intensity, while edges and gradients characterize the visible image's texture detail information. According to the different imaging characteristics of the source image, the selection of traditional manually designed fusion rules to represent the fused image, in the same way, will lead to the lack of diversity of extracted features, which may bring artifacts to the fused image. Moreover, for multi-source image fusion, manual fusion rules will make the method more and more complex. In view of the above problems, the image fusion method based on deep learning can assign weights to the model through an adaptive mechanism [34]. Compared with the design rules of traditional methods, this method greatly reduces the calculation cost, which is crucial in many fusion rules. Therefore, this research aims to conduct a detailed review of the existing deep learning-based infrared and visible image fusion algorithms and discuss their future development trends and challenges. Second, this article also introduces the theoretical knowledge of infrared and visible image fusion and the corresponding fusion evaluation index. This survey also makes a qualitative and quantitative comparison of some related articles' experiments to provide a reliable basis for this research. Finally, we summarized the fusion methods in recent years and analyzed future work trends.

The structure of this survey is schematically illustrated in Figure 1. Section 2 discusses the image fusion method based on deep learning. Section 3 provides an overview of the metrics used to measure fusion quality. In Section 4, we extensively experiment to compare the typical algorithms in each category and provide an objective performance. Section 5 discusses the future development trends and problems of infrared and visible image fusion. In Section 6, we have summarized the whole article.

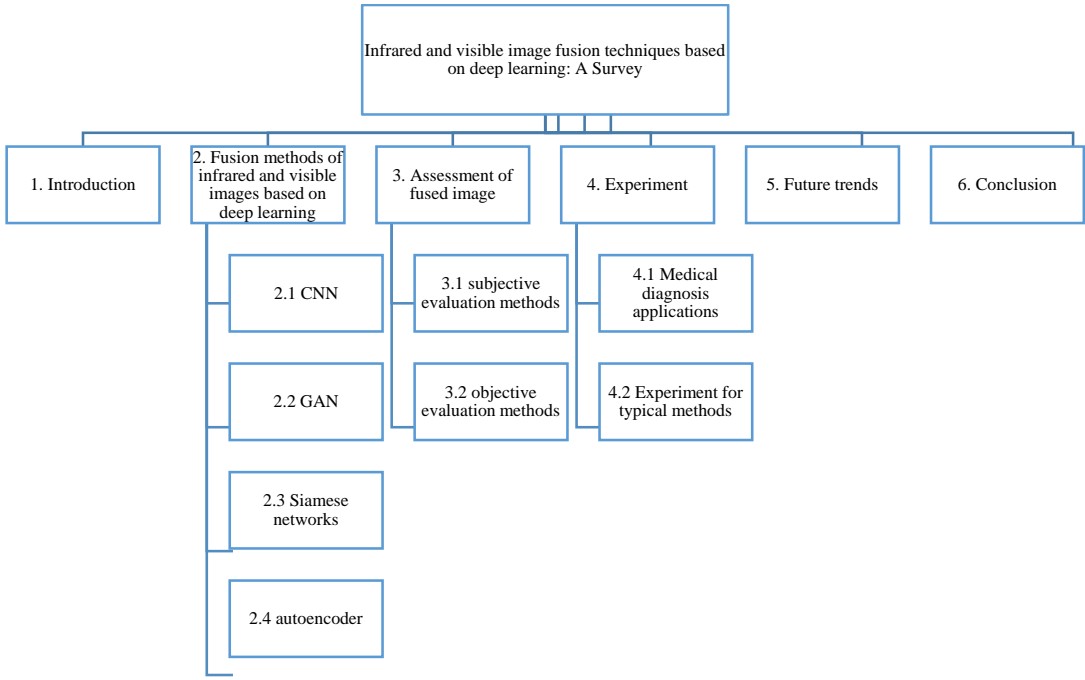

**Figure 1.** Structure of the survey.

## 2. Fusion Methods of Infrared and Visible Images Based on Deep Learning

In this section, we comprehensively review the infrared and visible image fusion methods based on deep learning. Increasing new methods of using deep learning for infrared and visible image fusion have been produced in recent years. These state-of-the-art methods are widely used in many applications, like image preprocessing, target recognition, and image classification. The traditional fusion framework can be roughly summarized in Figure 2. The two essential factors of these algorithms are feature extraction and feature fusion. Their main theoretical methods can be divided into multiscale transformation, sparse representation, subspace analysis, and hybrid methods. However, these artificially designed extraction methods make the image fusion problem more complicated due to their limitations. In order to overcome the limitations of traditional fusion methods, deep learning methods are introduced for feature extraction. In recent years, with the development of deep learning, several fusion methods based on convolutional neural network (CNN), generative adversarial networks (GAN), Siamese network, and autoencoder have appeared in the field of image fusion. The main fusion methods involved in this section are listed in Table 1 by category. Image fusion results based on deep learning have good performance, but many methods also have apparent challenges. Therefore, we will introduce the details of each method in detail.

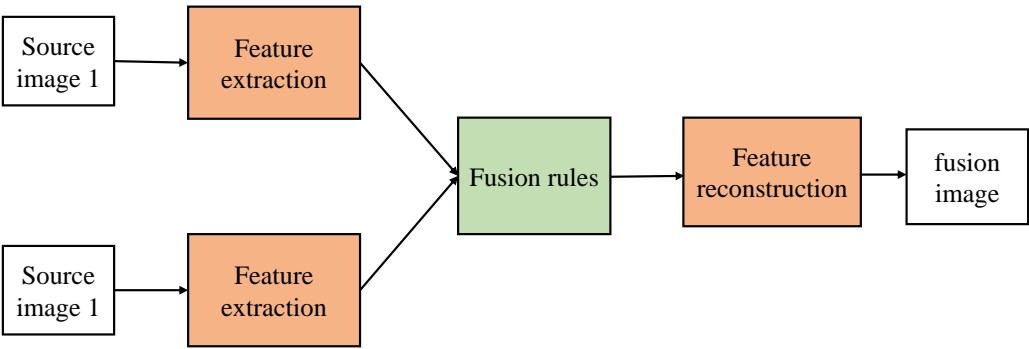

**Figure 2.** Traditional image fusion framework.

**Table 1.** The overview of some deep learning (DL)-based fusion methods.

| Families of Fusion Methods | Ref. | Innovation |
|---|---|---|
| CNN method of DL | [35] | VGG-19; L1 norm; weighted average strategy; maximum selection strategy |
| | [36] | Dense net |
| | [37] | Minimize the total change |
| | [38] | ZCA-zero-phase component analysis; L1-norm; weighted average strategy |
| | [39] | Elastic weight consolidation |
| | [40] | Perceptual loss; use two convolutional layers to extract image features; weight sharing strategy |
| | [41] | Adaptive information preservation strategy |
| | [42] | MLF-CNN; weighting summation strategy |
| | [43] | Mixed loss function (M-SSIM loss; TV loss); adaptive VIF-Net |
| Siamese network of DL | [44] | Fusion strategy of local similarity; weighted average |
| | [45] | Pixel-level image fusion; feature tracking |
| | [46] | Dual Siamese network; weight sharing strategy |
| | [47] | Saliency map; three-level wavelet transform |
| GAN of DL | [33] | The confrontation between generator and discriminator |
| | [48] | Learnable group convolution |
| | [49] | Adversarial generation network with dual discriminators |
| | [50] | Detail loss; target edge loss |
| | [51] | Local binary pattern |
| | [52] | Pre-fused image as the label |
| Autoencoder of DL | [49] | Automatic coding feature extraction strategy of generator |
| | [53] | Combination of autoencoder and dense network |
| | [54] | RGB encoder; infrared encoder; decoder used to restore the resolution of the feature map |

*2.1. CNN-Based Fusion Methods*

In computer vision, convolutional layers play an important role in feature extraction and usually provide more information than traditional manual feature extraction methods [55,56]. The critical problem of image fusion is how to extract salient features from the source images and combine them to generate the fused image. However, CNN has three main challenges when applied to image fusion. First, training a good network requires much labeled data. However, the image fusion architecture based on the convolutional neural network is too simple, and the convolutional calculation layer in the network framework is less, and the features extracted from the image are insufficient, resulting in poor fusion performance. Second, the artificially designed image fusion rules are challenging to realize the end-to-end model network, and some errors will be mixed in the feature reconstruction process, which will affect the feature reconstruction of the image. Finally, the efficient information of the last layer is ignored in the traditional convolutional neural network algorithm, so that the model features cannot be fully retained. With the deepening of the network, the feature loss will become severe, resulting in a worsening of the final fusion effect.

In [57], Liu et al. proposed a fusion method based on convolutional sparse representation (CSR). In their method, the authors use CSR to extract multilayer features and then use them to generate

fusion images. In [58], they also proposed a fusion method based on a convolutional neural network (CNN). They use image patches containing different feature inputs to train the network and obtain a decision graph. Finally, the fusion image is obtained by using the decision graph and the source image. Li et al. [35] proposed a simple and effective infrared and visible image fusion method based on a deep learning framework. The article divides the source image information into two parts, the former contains low-frequency information, and the latter contains texture information. The model is based on the multilayer fusion strategy of the VGG-19 network [59] through which the deep features of the detailed content can be obtained. In other multiple exposure fusion (MEF) algorithms, they rely on artificially searched features to fuse images. When the input conditions change, the parameters will follow the change, so the robustness of the algorithm cannot be guaranteed, and processing multiple exposure images will consume a lot. The learning ability of CNN is affected mainly by some loss functions. Prabhakar et al. [36], the proposed method does not need parameter adjustment when the input changes. The fusion network consists of three parts: the encoder, the fusion layer, and the decoder. To combine encoder networks employing encoders. From the perspective of the CNN method, by optimizing the parameters of the loss function learning model, the results can be predicted as accurately as possible. In [37], Ma et al. proposed an infrared and visible image based on the minimization of the total variation (TV) by limiting the fusion image to have similar pixel intensity to the infrared image and similar gradient to the visible image. In [38], Li et al. proposed a fusion framework based on deep features and zero-phase component analysis. First, the residual network is used to extract the depth features of the source image, and then the ZCA-zero-phase component analysis [60] and L1-norm are used for normalization to obtain the initial weight map. Finally, the weighted average strategy is used to reconstruct the fused image.

Xu et al. [39], a new unsupervised and unified densely connected network is proposed. The densely connected network (DenseNet) [61] is trained to generate a fused image adjusted on the source image in the proposed method. In addition, we obtain a single model applicable to multiple fusion tasks by applying elastic weight consolidation to avoid forgetting what has been learned from previous tasks when training multiple tasks sequentially, rather than train individual models for every fusion task or jointly train tasks roughly. The weight of the two source images is obtained through the weight block, and different feature information is retained. The model generates high-quality fusion results in processing multi-exposure and multi-focus image fusion. In [40], Zhang et al. proposed an end-to-end model divided into three modules: feature extraction module, feature fusion module, and feature reconstruction module. Two convolutional layers are used to extract image features. Appropriate fusion rules are adopted for the convolutional features of multiple input images. Finally, the fused features are reconstructed by two convolutional layers to form a fused image. In [41], Xu et al. believe that an unsupervised end-to-end fusion network can solve different fusion problems, including multimode, multi-exposure, and multi-focus. The model can automatically estimate the importance of the corresponding source image features and provide adaptive information preservation because the model has an adaptive ability to retain the similarity between the fusion result and the source image. It dramatically reduces the difficulty of applying deep learning to image fusion-the universality of the model and the adaptive ability of training weights. Solve the catastrophic forgetting problem and computational complexity.

In [42], Chen et al. used deep learning methods to fuse visible information and thermal radiation information in multispectral images. This method uses the multilayer fusion (MLF) area network in the image fusion stage. In this way, pedestrians can be detected at different ratios under unfavorable lighting (such as shadows, overexposure, or night) conditions. To be able to handle targets of various sizes, prevent the omission of some obscure pedestrian information. In the region extraction stage, MLF-CNN designed a multiscale region proposal network (RPN) [62] to fuse infrared and visible light information and use summation fusion to fuse two convolutional layers. In [43], to solve the lack of label dataset, Hou et al. used a mixed loss function. The thermal infrared image and the visible image were adaptively merged by redesigning the loss function, and noise interference was

suppressed. This method can retain salient features and texture details with no apparent artifacts and have high computational efficiency. We make an overview list of some of the image fusion based on CNN in Table 2.

**Table 2.** The overview of some a convolutional neural network (CNN)-based fusion methods.

| Ref. | Limitation |
|:---:|:---|
| [57] | It is only suitable for multi-focus image fusion, and only the last layer is used to calculate the result. Much useful information obtained by the middle layer will be lost. When the network depth increases, the information loss will become more serious. |
| [36] | Feature extraction will still lose some information. |
| [37] | In different application fields, the accuracy of the fusion result cannot be guaranteed due to the large difference in resolution and spectrum. |
| [40] | The specific performance of different source images needs to be considered in a specific dataset. |
| [42] | A large number of samples with a complex background bring a large amount of calculation to model training. |

## 2.2. Siamese Networks-Based Fusion Methods

Part of the difficulty of image fusion is that infrared images and visible images have different imaging methods. In order to make the fusion image retain the relatively complete information of the two source images at the same time, a pyramid framework is used to extract feature information from the infrared image and the visible image, respectively.

Liu et al. [58] recently proposed a Siamese convolutional network, especially image fusion. The network input is two source images, while the output is a weight map for the final decision. Many high-quality natural images are applied to generate the training dataset via Gaussian blurring and random sampling. The main characteristic of this approach is activity level measurement, and weight assignments are simultaneously achieved with the network. In particular, the convolutional layers and fully-connected layers could be viewed as the activity level measurement and weight assignment parts in image fusion, respectively. Again in [44], Liu et al. proposed a convolutional neural network-based infrared and visible image fusion method. This method uses the Siamese network to obtain the network weight map. The weight map combines the pixel activity information of the two source images. The model has mainly divided into four steps: the infrared image and the visible image are passed into the convolutional neural network to generate weights; the Gaussian pyramid is used to decompose the weight of the source image, and the two source images are decomposed by the Laplacian pyramid respectively. The information obtained by the decomposition of each pyramid is fused with coefficients in a weighted average manner. Figure 3 clearly explains the working principle of the Siamese network in the fusion process. In [45], Zhang et al. believe that CNN has a powerful feature representation ability and can produce good tracking performance. Still, the training and updating of the CNN model are time-consuming. Therefore, in this paper, the Siamese network is used for pixel-level fusion to reduce time consumption. First, the infrared and visible images are fused and then put into the Siamese network for feature tracking. In [46], Zhang et al. used a fully convolutional Siamese network fusion tracking method. SIamFT uses a Siamese network, a visible light network, an infrared network. They are used to process visible and infrared images, respectively. The backbone uses the SiamFC network, the visible light part of the network weight sharing, and the infrared part of the network weight sharing. The operating speed is about [35–38,58] FPS so that it can meet real-time requirements. In [47], Piao et al. designed an adaptive learning model based on the Siamese network, which automatically generates the corresponding weight map through the saliency of each pixel in the source image to reduce the number of traditional fusion rules. The parameter redundancy problem. This paper uses a three-level wavelet transform to decompose the source image into a low-frequency weight map and a high-frequency weight map. The scaled weight map is used to reconstruct the wavelet image to obtain the corresponding fused image. This result is more consistent with the human

visual perception system. There are fewer undesirable artifacts. We make an overview list of some image fusion based on the Siamese network in Table 3.

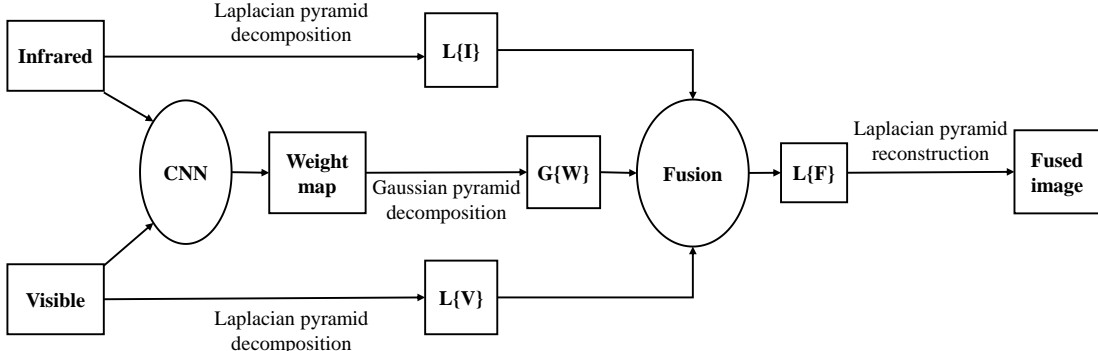

**Figure 3.** Siamese network-based infrared and visible image fusion scheme (credit to [44]).

**Table 3.** The overview of Siamese network-based fusion methods.

| Ref. | Limitation |
|------|------------|
| [45] | The starting point of the article is target tracking. As far as the fusion effect is concerned, the fusion result is slightly blurred. |
| [44] | It cannot be effectively combined with conventional fusion technology and is not suitable for complex data sets. |
| [46] | The thermal infrared network training uses visible images, and you can consider using thermal infrared images for better results. |
| [47] | The CPU is used to train the model, so the computational efficiency of the model is not very prominent. It takes an average of 19 s to process a pair of source images. |

### 2.3. GAN-Based Fusion Methods

The existing deep learning-based image fusion technology usually relies on the CNN model, but in this case, the ground truth needs to be provided for the model. However, in the fusion of infrared and visible images, it is unrealistic to define fusion image standards. Therefore, without considering the ground truth, a deep model is learned to determine the degree of blurring of each patch in the source image, and then the weight is calculated. Map accordingly to generate the final fusion image [44]. Using a generative countermeasure network to fuse infrared and visible images can be free from the above problems.

In [33], Ma et al. proposed an image fusion method based on a generative confrontation network, where the generator is mainly for the fusion of infrared images and visible images, and the purpose of the discriminator is to make the fused image have more details in the visible image, which makes the fused image. The infrared heat radiation information and visible texture information can be kept in the fusion image simultaneously. Figure 4 shows the image fusion framework based on GAN. For fusion GAN, the source image's vital information cannot be retained at the same time during the image fusion process, and too much calculation space is occupied during the convolution process. In [48], learning group convolution is used to improve the efficiency of the model and save computing resources. In this way, a better tradeoff can be made between model accuracy and speed. Moreover, the remaining dense blocks are used as the fundamental network construction unit. The inactive perceptual characteristics are used as the input content loss characteristics, which achieves deep network supervision.



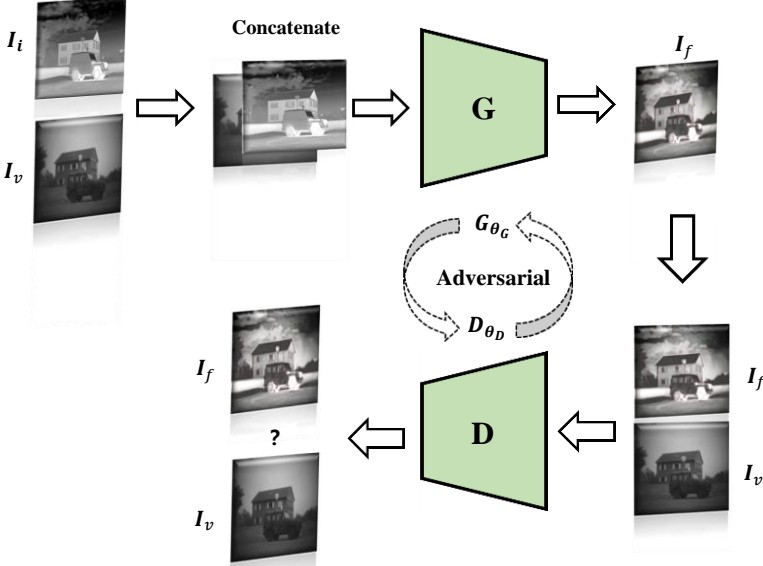

**Figure 4.** GAN-based infrared and visible image fusion framework.

In [49], Ma et al. make the fusion image similar to the infrared image by constrained sampling to avoid blurring radiation information or loss of visible texture details. The dual discriminator does not need ground truth fusion images for pre-training, which can fuse images of different resolutions without causing thermal radiation information blur or visible texture detail loss. Considering the two challenges of CNN, relying only on adversarial training will result in the loss of detailed information. Therefore, a minimax game is established between the generator and the discriminator in [50]. The loss of the model becomes the loss of detail, the loss of the target edge, and confrontation loss. In [51], Xu et al., based on local binary pattern (LBP) [63], intuitively reflected the edge information of the image by comparing the values between the central pixel and the surrounding eight pixels to generate a fusion image with richer boundary information. The discriminator encodes and decodes the fused image and each source image, respectively, and measures the difference between the distributions after decoding. In [52], Li et al. used the pre-fused image as the label strategy so that the generator takes the pre-fused image as the benchmark in the generation process so that the image fused by the generator can effectively and permanently retain the rich texture in the visible image and the thermal radiation information in the infrared image. We make an overview list of some of the image fusion based on GAN in Table 4.

**Table 4.** The overview of some GAN-based fusion methods.

| Ref. | Limitation |
| --- | --- |
| [33] | Reduce the prominence of infrared thermal targets. |
| [50] | The pixel intensity of some fusion image areas is changed, and the overall brightness is reduced. |
| [48] | Some edges of the fused image are a bit blurry. |
| [51] | Unique fusion results have bright artifacts. |
| [52] | In the early stage of model training, it takes some time to label the pre-fused images. |

*2.4. Autoencoder-Based Fusion Methods*

In the paper [36], Prabhakaret et al. studied the fusion problem based on CNN. They proposed a simple CNN-based architecture, including two encoding network layers and three decoding network layers. Although this method has good performance, there are still two main shortcomings: (1) The network architecture is too simple, and it may not be able to extract the salient features of the source image correctly; (2) These methods only use the last layer of the encoding network to calculate; as a

result, the useful information obtained by the middle layer will be lost. This phenomenon will become sparser when the network is deeper. In the traditional CNN network, as the depth increases, the fusion ability of the model is degraded [30]. For this problem, Heet et al. [64] introduced a deep residual learning framework to improve the layers' information flow further. Huang et al. [61] proposed new architecture with dense blocks in which each layer can be directly connected to any subsequent layer. The main advantages of the dense block architecture: (1) the architecture can retain as much information as possible; (2) the model can improve the information flow and gradient through the network, and the network is easy to train; (3) this dense connection method has a regularization effect, which can reduce overfitting caused by too many parameters [61].

In [53], Li et al. combine the encoding network with the convolutional layer, fusion layer, and dense block, and the output of each layer is connected. The figure shows the working principle of the Autoencoder model in the fusion image. The model first obtains the feature map through CNN and dense block and then fuses the feature through the fusion strategy. After the fusion layer, the feature map is integrated into a feature map containing the significant features of the source image. Finally, the fused image is reconstructed by a decoder. The fusion mechanism of the autoencoder is shown in Figure 5. In [49], Ma et al. considering the existing methods to solve the difference between output and target by designing loss function. These indicators will introduce new problems. It is necessary to design an adaptive loss function to avoid the ambiguity of the results. Most human-designed fusion rules lead to the extraction of the same features for different types of source images, making this method unsuitable for multi-source image fusion. In this paper, a double discriminator is used to pre-train the fused images. An Autoencoder is used to fuse the images with different resolutions to retain the maximum or approximately the maximum amount of information in the source images. In [54], Sun et al. used the RGB-thermal fusion network (RTFNet). RTFNet consists of three modules: RGB encoder and infrared encoder for extracting features from RGB images and Thermal images, respectively, and decoder to restore the resolution of feature images. Where the encoder and decoder are designed regionally symmetric, RTFNet is used for feature extraction, where the new encoder can restore the resolution of the approximate feature map. As this method is mainly used for scene segmentation, the edge of scene segmentation is not sharp.

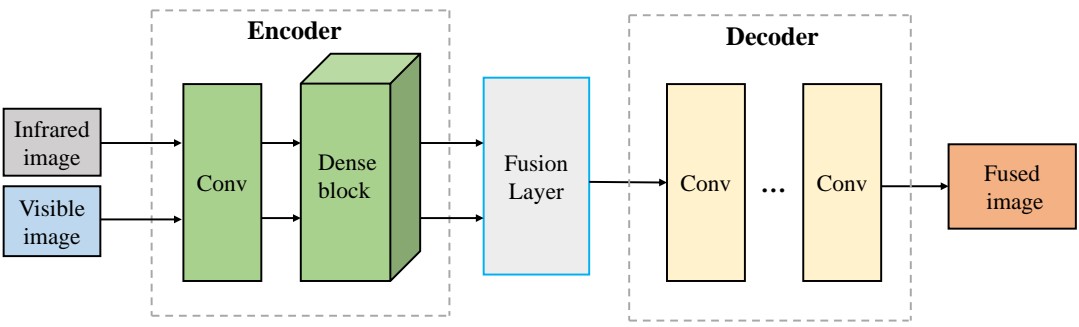

**Figure 5.** Autoencoder based infrared and visible image fusion framework (credit to [53]).

## 3. Assessment of Fused Image

Infrared and visible light image fusion has attracted widespread attention in the field of image fusion. Due to the advancement of image fusion technology, it is necessary to evaluate several related image fusion methods that have been proposed. However, different fusion methods have different characteristics, and there are significant differences in the actual application process. Many researchers use fusion evaluation indicators to evaluate fused infrared and visible images [65]. The existing integration indicators are roughly divided into subjective evaluation and objective evaluation [66]. Objective evaluation can avoid the above problems. Researchers can use different point-specific evaluation indicators for quantitative reference and make accurate comparisons of image fusion.

## 3.1. Subjective Evaluation Methods

The subjective evaluation can be further divided into the absolute evaluation and relative evaluation, which is implemented by a widely recognized five-level quality scale and obstacle scale [67]. The most direct form of subjective evaluation is human eye observation. Humans compare different fusion methods by observing image details, contrast, image integrity, and degree of distortion. Subjective evaluators can use the evaluation criteria to score qualitatively on the fused image directly. Still, everyone has different evaluation criteria for the same image, which is easily affected by personal preference, environment, and other factors, leading to incorrect responses to the fused image. This method's quality is poor, and the timeliness is low, which is not conducive to a multidimensional evaluation of fusion images. It is necessary to combine objective evaluation indicators to evaluate the fusion results reasonably.

## 3.2. Objective Evaluation Metrics

Objective evaluation can avoid problems existing in subjective evaluation. Researchers can make quantitative reference for evaluation indicators with different emphases and make an accurate comparison for image fusion. There are many kinds of objective fusion indicators, which are defined according to some computable mathematical formulas. The objective evaluation is to establish a corresponding mathematical model by simulating the human visual perception system and compare the performance of each fusion method through statistical data. Objective image quality measurement includes the use of reference standard images and non-reference standard images. Common objective evaluation indicators have been listed in Table 5. In this section, we mainly introduce the relevant evaluation indicators in the table in detail.

**Table 5.** Statistics of some representative fusion evaluation measures and references. "+" means that a large value indicates good performance, while "−" means that a small value indicates good performance.

| Evaluation Metrics | References | +/− |
|---|---|---|
| Entropy (EN) | [68–72] | + |
| Spatial frequency (SF) | [73–79] | + |
| Similarity (COSIN) | [77] | + |
| Correlation coefficient (CC) | [59,80–82] | + |
| Standard deviation (SD) | [31,83–85] | + |
| Structural similarity index measure (SSIM) | [86–88] | + |
| Mutual information (MI) | [89–92] | + |
| Average gradient (AG) | [73,93–95] | + |
| Mean squared error (MSE) | [96–98] | − |
| Gradient-based fusion performance (QAB/F) | [99–103] | + |
| Peak signal-to-noise ratio (PSNR) | [2,104–107] | + |
| Visual information fidelity of fusion (VIFF) | [108] | + |
| Chen–Blum metric (QCB) | [66] | + |
| Chen–Varshney metric (QCV) | [109] | − |

### 3.2.1. Entropy (EN)

EN is a form of statistical features that can reflect the average amount of information in an image. Its mathematical definition is as follows:

$$EN = -\sum_{l=0}^{L-1} p_l log_2 p_l,\qquad(1)$$

according to Shannon's method of using information entropy as the quantification of information content [110], where $l$ represents the number of gray levels of the image, $P_l$ represents the proportion of pixels with a gray value of 1 in the total pixels. The higher the *EN*, the richer the information and the better the quality of the fused image.

### 3.2.2. Spatial Frequency (SF)

The spatial frequency *SF* is based on the image gradient to reflect the image detail and texture sharpness level. *SF* can be divided into spatial row frequency *RF* and spatial column frequency *CF*. Its mathematical definition is as follows:

$$F = \sqrt{RF^2 + CF^2}, \tag{2}$$

$$RF = \sqrt{\frac{1}{M \times N} \sum_{i=1}^{M} \sum_{j=2}^{N} [P(i,j) - P(i,j-1)]^2}, \tag{3}$$

$$CF = \sqrt{\frac{1}{M \times N} \sum_{i=2}^{M} \sum_{j=1}^{N} [P(i,j) - P(i-1,j)]^2}, \tag{4}$$

where *P* indicates that the final fusion image size is *M* × *N*. First, the horizontal frequency *RF* and vertical frequency *CF* are calculated, and then the *SF* is calculated. The larger the *SF* value, the richer the edge texture information, and the more in line with the human visual perception system.

### 3.2.3. Similarity (COSIN)

COSIN converts the corresponding image into a vector and calculates the cosine similarity between the vectors, and the mathematical formula of COSIN is defined as follows:

$$similarity = \cos(\theta) = \frac{A \cdot B}{\| A \| \| B \|} = \frac{\sum_{i=1}^{n} A_i \times B_i}{\sqrt{\sum_{i=1}^{n} (A_i)^2} \times \sqrt{\sum_{i=1}^{n} (B_i)^2}}, \tag{5}$$

This method has a large amount of calculation, while the result is more reliable than *SSIM*. The larger the *COSIN* value, the more similar the pixels in the image are.

### 3.2.4. Correlation Coefficient (CC)

The CC measures the degree of linear correlation between a fused image and infrared and visible images and is mathematically defined as follows:

$$CC = \frac{(r_{I,F} + r_{V,F})}{2}, \tag{6}$$

$$r_{X,F} = \frac{\sum_{i=1}^{H} \sum_{j=1}^{W} \left( X(i,j) - \overline{X} \right) \left( F(i,j) - \overline{F} \right)}{\sqrt{\sum_{i=1}^{H} \sum_{j=1}^{W} \left( X(i,j) - \overline{X} \right)^2 \left( \sum_{i=1}^{H} \sum_{j=1}^{W} \left( F(i,j) - \overline{F} \right)^2 \right)}}, \tag{7}$$

where *X* represents IR or VIS image. $\overline{X}$ and $\overline{F}$ denotes the source image and fused image *F* average pixel values, *H* and *W* stand for the length and width of the test image. The larger the fusion image of *CC* is closely related to source images, and the better the fusion performance.

### 3.2.5. Standard Deviation (SD)

SD is an objective evaluation index to measure the richness of image information. Standard deviation is used to measure the change of pixel intensity in the fused image. Its mathematical definition is as follows:

$$SD = \sqrt{\frac{1}{MN} \sum_{i=1}^{M} \sum_{j=2}^{N} \left[ H(i,j) - \overline{H} \right]^2}, \tag{8}$$

$$\overline{H} = \frac{1}{MN} \sum_{i=1}^{M} \sum_{j=2}^{N} H(i,j), \tag{9}$$

where $\overline{H}$ represents the mean value and can also be used to evaluate fusion images. The difference in standard deviation can reflect the difference in image contrast, which is similar to the high sensitivity of the human visual system to high-contrast areas. A high-contrast fusion image will produce a larger *SD*, which means that the fusion image has a clear contrast [75].

### 3.2.6. Structural Similarity Index Measure (SSIM)

SSIM models image loss and distortion [82], compares and measures the similarity in three aspects of image brightness, contrast, and structure, and finally combines the three components to produce an overall similarity measure. According to the change of image structure information, the image's distortion in three aspects is considered an objective evaluation. The mathematical definition is as follows:

$$SSIM(X, Y) = \left( \frac{2\mu_x\mu_y + c_1}{\mu_x^2 + \mu_y^2 + c_1} \right)^{\alpha} \cdot \left( \frac{2\sigma_x\sigma_y + c_2}{\sigma_x^2 + \sigma_y^2 + c_2} \right)^{\beta} \cdot \left( \frac{\sigma_{xy} + c_3}{\sigma_x\sigma_y + c_3} \right)^{\gamma}, \tag{10}$$

$$\sigma_{xy} = \frac{1}{N-1} \sum_{i=1}^{N} (x_i - \mu_x)(y_i - \mu_y), \tag{11}$$

where $x$ and $y$ are the reference image and fused image, respectively, $\mu_x$, $\mu_y$, $\sigma_x^2$, $\sigma_y^2$, and $\sigma_{xy}$ represent the mean value and variance and covariance of images $x$ and $y$, respectively, and $c_1$, $c_2$, and $c_3$ are small normal numbers to avoid having a denominator of zero. When $c_1 = c_2 = c_3 = 0$, *SSIM* will become a general fusion index [83]. Parameters $\alpha$, $\beta$, $\gamma$ are used to adjust the proportions. The value of *SSIM* will be between [−1, 1], and the larger the value, the more similar the fused image and the source image in terms of brightness, contrast, and structure.

### 3.2.7. Mutual Information (MI)

MI can measure the amount of information transmitted from the source image to the fusion image, how much information the fusion image has acquired from the original image [85]. *MI* reflects the statistical dependence of two random variables from the perspective of information theory, and its mathematical definition is:

$$MI_F^{AB} = MI_{FA} + MI_{FB}, \tag{12}$$

where $MI_{FA}$ and $MI_{FB}$ represent the amount of information from infrared images and visible images to fusion images, a large *MI* metric means that considerable information is transferred from source images to the fused image, which indicates a good fusion performance.

### 3.2.8. Average Gradient (AG)

*AG* can represent the ability to express the texture and detail of the fused image and can be used to evaluate the sharpness of the image. Its mathematical definition is:

$$AG = \frac{1}{M \times N} \sum_{i=1}^{M} \sum_{j=1}^{N} \sqrt{\frac{1}{2}((F(i, j) - F(i+1, j))^2 + (F(i, j) - F(i, j+1))^2)} \tag{13}$$

The final fusion image size is $M \times N$, and $F(i, j)$ represents the grayscale pixels of the image at the pixel level $(i, j)$. Generally speaking, the larger the average gradient value, the richer the information in the fusion image and the better the fusion effect.

### 3.2.9. Mean Squared Error (MSE)

*MSE* calculates the error between the fused image and the source image to measure the difference between the two. Its mathematical definition is:

$$MSE = \frac{MSE_{AF} + MSE_{BF}}{2}, \tag{14}$$

where $MSE_{AF}$ and $MSE_{BF}$ represent the differences between fused images and infrared and visible images, respectively. The *MSE* calculation method between the fusion image and each source image is as follows:

$$MSE_{XF} = \frac{1}{MN} \sum_{l=0}^{M-1} \sum_{j=0}^{N-1} (X(i,j) - F(i,j))^2 \tag{15}$$

The smaller the *MSE*, the smaller the difference between the fused image and the source image, and the better the fusion performance.

### 3.2.10. Gradient-Based Fusion Performance ($Q^{AB/F}$)

$Q^{AB/F}$ uses local metrics to estimate the degree of retention of the source image's edge information in the fusion image. Its mathematical definition is:

$$Q^{AB/F} = \frac{\sum_{n=1}^{N} \sum_{m=1}^{M} Q^{AF}(n,m) w_A(n,m)}{\sum_{i=1}^{N} \sum_{j=1}^{M} (w_A(i,j) + w_B(i,j))}, \tag{16}$$

among them, $Q^{XF}(n,m)$ is calculated as follows:

$$Q^{XF}(n,m) = Q_g^{XF}(n,m) Q_a^{XF}(n,m), \tag{17}$$

where $Q_g^{XF}(n,m)$ and $Q_a^{XF}(n,m)$ denote the edge strength and orientation values at the location $(n,m)$, $W_X$ denotes the weight that expresses the importance of each source image to the fused image. The larger the $Q^{AB/F}$ value is, the more edge information in the source image is retained in the fused image and the better fusion effect is achieved. The dynamic range of $Q^{AB/F}$ is [0, 1]. The closer the value is to 0, the more edge information is lost. On the contrary, the closer to 1 means that more information about the source image is retained.

### 3.2.11. Peak Signal-to-Noise Ration (PSNR)

*PSNR* can calculate the ratio of peak power and noise power in the fusion image, and its mathematical definition is:

$$PSNR = 10 \log_{10} \frac{r^2}{MSE}, \tag{18}$$

where *r* represents the peak value of the fused image, and *MSE* represents the mean square error. *PSNR* can reflect the distortion degree of the fusion image. The higher the value is, the closer the fusion image is to the source image, the better the fusion effect will be.

### 3.2.12. Visual Information Fidelity of Fusion (VIFF)

VIFF is only used in image fusion and is developed based on visual information fidelity (VIF) [111]. Yu et al. [108] used the VIF model to extract the visual information of the source image. They obtained the effective fusion of visual information after further processing to remove the distortion information. Finally, after integrating all the visual information, the VIFF specially used for fusion evaluation is obtained. According to [108], the calculation process of VIFF is summarized into four parts. First, filter the source image and the fusion image and divide them into different blocks. Second, evaluate whether each block has distorted visual information. Third, calculate the fidelity of the visual information of each block. Fourth, the overall index based on VIF is calculated.

### 3.2.13. Other Metrics

$Q_{CB}$ and $Q_{CV}$ are indicators inspired by human perception and used to measure the visual performance of fused images. Running time is an essential indicator for evaluating the performance of

an algorithm. The time complexity of an image fusion algorithm is used to evaluate the computational efficiency of the model.

## 4. Experiments

The feature information of infrared and visible images has many complementary features so that the fusion algorithm can be embedded in many applications and improve the original method [7]. The fusion technology of infrared and visible images has been applied to target detection [8], tracking [107], surveillance [18], remote sensing [102], and medical image processing [82]. At present, there are several existing visible and infrared image fusion data sets, including OSU color thermal database [112], TNO image fusion dataset [113], INO video analytics dataset, VLIRVDIF [114], VIFB [115], and RGB-NIR Scene dataset [116]. Table 6 summarizes the preliminary information about these datasets. In fact, except for OSU, there are not many image pairs in TNO and VLIRVDIF. However, the lack of a code library, evaluation metrics, and the results of these datasets make it difficult to evaluate the latest technologies against them. This section first introduces the fusion method in the treatment of complex color images of the space application. Second, we aimed at several specific infrared and visible image fusion methods that were evaluated and had a comprehensive understanding of their practical application characteristics and performance.

**Table 6.** Details of existing infrared and visible image fusion datasets.

| Name | Image/Video Pairs | Image Type | Resolution | Year |
|---|---|---|---|---|
| TNO | 63 image pairs | Multispectral | Various | 2014 |
| VLRVDIF | 24 video pairs | RGB, infrared | $720 \times 480$ | 2019 |
| OSU color-thermal database | 6 video pairs | RGB, infrared | $320 \times 240$ | 2005 |
| VIFB | 21 image pairs | RGB, infrared | Various | 2020 |
| INO-video analytics dataset | 223 image pairs | RGB, infrared | Various | 2012 |
| RGB-NIR scene dataset | 477 image pairs | RGB, near-infrared (NIR) | $1024 \times 768$ | 2011 |

### 4.1. Medical Diagnosis Applications

Medical image fusion aims to improve image quality by retaining specific features, increasing the applicability of the image in clinical diagnosis, and evaluating medical problems [82]. With the rapid development of sensors and image fusion, medical image fusion has played a vital role in various clinical applications, including medical diagnosis [104], surgical navigation, and treatment planning. It is an essential tool for doctors to diagnose diseases accurately [103]. In medical imaging, X-ray, magnetic resonance imaging (MRI), and computed tomography are typical structural systems. MRI images are similar to visible images. It is excellent in capturing the details of the soft tissue structure of the brain, heart, and lungs with high-resolution. Positron emission tomography (PET) images are similar to infrared images in that they are obtained through nuclear medicine imaging and can provide functional and metabolic information, such as blood flow activity. However, PET and single-photon emission computed tomography are usually rich in color, but low-resolution imaging limits their clinical applications due to limited resolution. Therefore, by fusing these two types of medical images, the result will contain the spatial and spectral characteristics of the source image [49]. As shown in Figure 6a, PET images are usually regarded as color images, and color images represent useful information. To retain useful information, the color of the fusion image should be as similar to the color of the PET image as possible. To preserve the image color in the image fusion, we adopt the PET image to the $YC_bC_r$ color space and only fuse the Y channel. The Y channel refers to the luminance component, $C_b$ refers to the blue chrominance component, and $C_r$ refers to the red chrominance component. The human eye is more sensitive to the Y component of the image, so after subsampling the chrominance component $C_b$ or $C_r$ to reduce the chrominance component, the naked eye will not perceive a significant change in image quality [117]. Therefore, in the experiment process,

as shown in Figure 7, we first converted the PET image into Y, $C_b$, and $C_r$ channels through $YC_bC_r$ and only fused the MRI image with the Y channel. Then the fused image is fused with $C_b$ and $C_r$ color components and inversely transformed to RGB space to obtain the final fusion result.

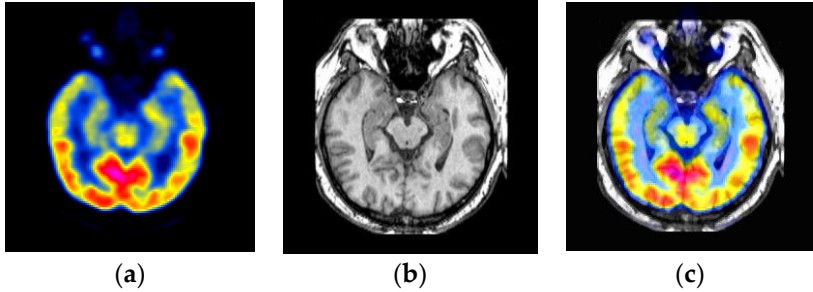

| (a) | (b) | (c) |

**Figure 6.** Examples of medical image fusion. From left to right: (**a**) PET image, (**b**) MRI image, (**c**) fused image.

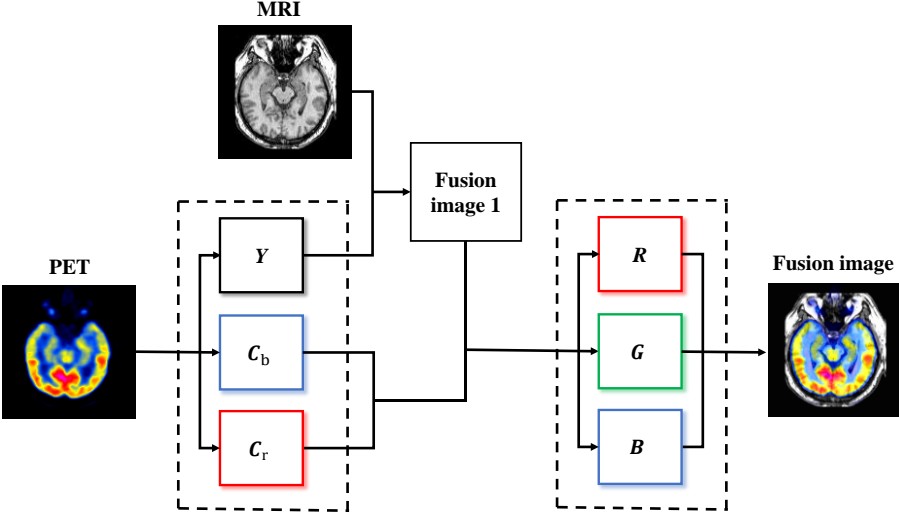

**Figure 7.** The fusion of PET and MRI images.

*4.2. Experiment for Typical Methods*

This section has commonly used infrared and visible images for qualitative evaluation experiments for several typical fusion methods, as shown in Figure 8. To evaluate different image fusion methods' performance, we used the seven most commonly used evaluation indicators, namely SF, EN, CC, COSIN, SD, SSIM, MI, and $Q^{AB/F,}$ to evaluate the fusion results quantitatively. The experiments are conducted with 3.7 GHz Intel 10,900×, GPU RTX 2080TI, and 11 GB memory.

It can be seen from the qualitative experimental results that the fusion results of the guided filtering based fusion method (GFF) method have a large area of artifacts, and the thermal radiation target is not prominent. However, the fusion images of dual-discriminator conditional generative adversarial network (DDcGAN), fully learnable group convolution (FLGC)-fusion GAN, and fusion GAN have more apparent texture details than other images, higher image contrast, and more prominent thermal radiation targets. The gradient transfer fusion (GTF) method has a good fusion effect on Bunker, Kaptein_1123, Kaptein_1654, and Lake images. However, when applied to the scene of Sandpath, it is easy to find that the thermal radiation targets in the image are not prominent. Therefore, from the perspective of qualitative analysis, most of the above methods need further optimization.

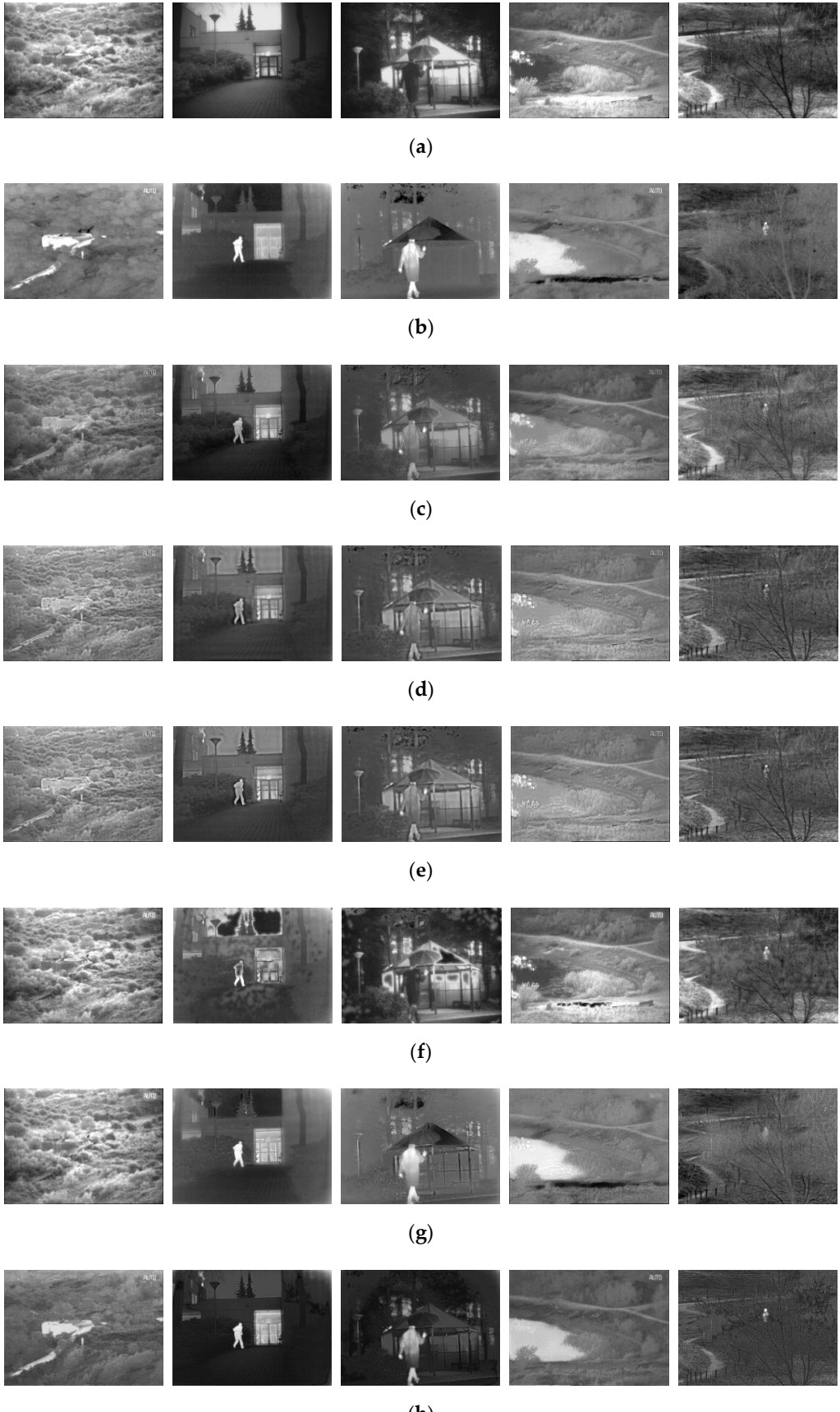

**Figure 8.** *Cont.*

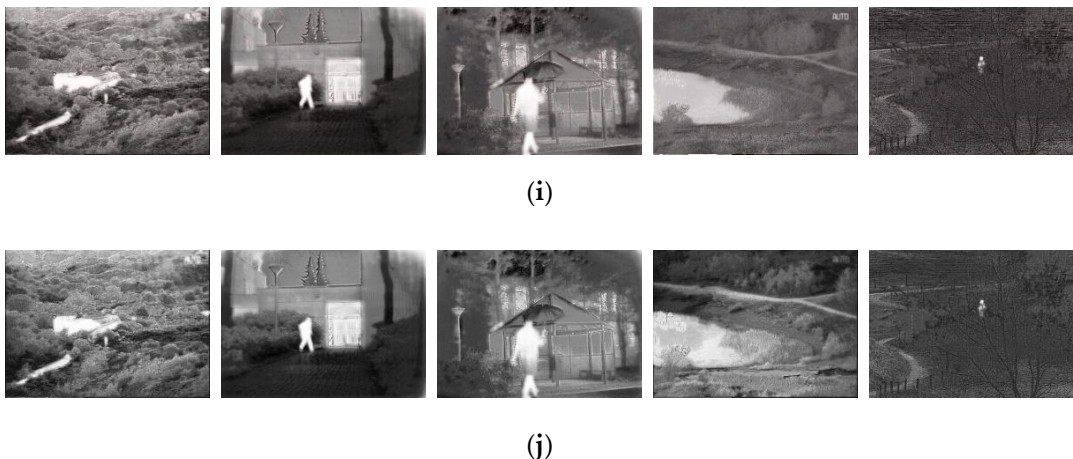

**Figure 8.** Qualitative fusion results on five typical infrared and visible image pairs from the TNO database. From left to right: Bunker, Kaptein_1123, Kaptein_1654, Lake, and Sandpath. From top to bottom: (**a**) the visible image, (**b**) the infrared image, (**c**) deep fuse [36], (**d**) curvelet transform (CVT) [89], (**e**) dual-tree complex wavelet transform (DTCWT) [118], (**f**) GFF [26], (**g**) gradient transfer fusion (GTF) [101], (**h**) fusion GAN [33], (**i**) FLGC-fusion GAN [48], (**j**) DDcGAN) [49].

We further use the above seven fusion indicators to conduct a quantitative comparison between ten pairs of infrared images and visible images in the TNO dataset, as shown in Table 7. Quantitative experimental results show that each fusion algorithm has advantages and disadvantages, and different methods show different advantages in different aspects. With deep learning in infrared and visible image fusion, superior new technologies have been continuously emerging to achieve better fusion results. In general, the fusion effect of GTF, fusion GAN, FLGC-fusion GAN, and DDcGAN is better than other methods in terms of brightness, texture detail, and contrast. It can be seen from Table 7 that DDcGAN with FLGC—fusion GAN was generally, even on individual indexes of several fusion methods, is better than the last. However, through Table 8, it is found that the operation efficiency of DDcGAN is lower than fusion GAN. Considering that the balance between computational complexity and fusion effect is essential for the fusion of infrared and visible images, infrared and visible image features will become a difficult and challenging problem. Most image fusion indicators can only reflect the quality of the fused image to a certain extent. Hence, it is necessary to study more effective fusion methods and evaluation indicators to conduct a comprehensive quality evaluation.

We summarized the running time of the eight fusion methods used for qualitative evaluation, as shown in Table 8. Although there are many new research methods in recent years, they are limited by a code library. Therefore, we evaluate some open-source fusion methods on the TNO dataset. For the experimental results of other fusion methods, please refer to related papers.

**Table 7.** The average evaluation index value with different methods.

| Methods | SF | EN | CC | COSIN | SD | SSIM | QAB/F |
|---|---|---|---|---|---|---|---|
| Deep Fuse | 6.9603 | 6.7448 | 0.4182 | 0.9706 | 44.0915 | 0.4259 | 0.4396 |
| CVT | 6.9696 | 6.2151 | 0.4023 | 0.9118 | 28.7006 | 0.394 | 0.5065 |
| DTCWT | 6.9266 | 6.6617 | 0.3892 | 0.9549 | 32.6006 | 0.4652 | 0.5359 |
| GFF | 7.0902 | 7.1238 | 0.3711 | 0.8683 | 43.2243 | 0.3763 | 0.6168 |
| GTF | 7.2161 | 7.1464 | 0.3622 | 0.9739 | 44.3197 | 0.3858 | 0.4154 |
| Fusion GAN | 7.4707 | 7.377 | 0.4473 | 0.9715 | 48.5708 | 0.4132 | 0.6237 |
| FLGC-fusion GAN | 8.0068 | 7.5199 | 0.4861 | 0.9801 | 50.6215 | 0.4541 | 0.6941 |
| DDcGAN | 7.9142 | 7.4589 | 0.5558 | 0.9847 | 50.6318 | 0.5093 | 0.6434 |

**Table 8.** Run time comparison of six methods on the TNO datasets. The fusion GAN, FLGC-fusion GAN, and DDcGAN are performed on graphics processing unit (GPU), while others are performed on the central processing unit (CPU). Each value denotes the mean of the run time of a specific method on a dataset (unit: second).

| Method | TNO |
| --- | --- |
| CVT | 1.46 |
| DTCWT | $3.32 \times 10^{-1}$ |
| GFF | $3.18 \times 10^{-1}$ |
| Deep Fuse | $4.27 \times 10^{-2}$ |
| GTF | 4.82 |
| Fusion GAN | $4.61 \times 10^{-2}$ |
| FLGC-fusion GAN | $3.07 \times 10^{-2}$ |
| DDcGAN | $5.19 \times 10^{-2}$ |

## 5. Future Trends

Image fusion technology has made certain achievements, but many difficulties have not found a perfect solution. For example, severe problems of visible spectrum detail information loss, real-time algorithm problems, imperfect evaluation system, etc. Among them, the loss of visible image information will directly lead to image fusion failure, and real-time performance is essential to the military reconnaissance-related fields where fusion technology is most widely used. The problems of image fusion need to be improved and researched in the future.

- At present, many image fusion models based on the convolutional neural network have good performance, but most of them are not perfect. For example, in order to retain the infrared thermal radiation information of the source image and the texture feature information of the visible image, the pixel intensity of the image is reduced;
- The fusion method based on the convolution neural network should pay attention to enhancing the liquidity of the middle layer network's features to fully retain the feature details extracted from each layer of convolution. In line with the future trend of unsupervised development, it can avoid human factors affecting image fusion performance;
- The existing fusion method generally has a large amount of calculation, and it is challenging to meet the real-time requirements. There are still many shortcomings for multi-source image registration;
- Different fusion methods have their advantages and disadvantages and should be universal for different application scenarios;
- In the future, different fields can be combined, such as image super-resolution, image denoising, and image fusion, and it is not limited to traditional fusion mechanisms;
- There is no standard fusion evaluation index in the field of image fusion. There are still some defects in the existing image fusion performance evaluation, such as image resolution, complex background environment, computational complexity. There is a lack of code libraries and benchmarks that can gauge the state-of-the-art.

## 6. Conclusions

The application of DL-based techniques to visible and infrared image fusion has been progressing at a fast rate in recent years. However, due to the complexity of application scenarios and the pursuit of computational efficiency and fusion effect, different applications of IR and VI image fusion still need to be further improved, and there are also potential development directions. This paper reviews the latest developments in DL-based image fusion technology and summarizes the issues that should be improved in this field in the future. This review investigates infrared and visible image fusion methods based on DL in recent years. These methods are mainly divided into four categories: CNN-based methods, GAN-based methods, Siamese network-based methods, and Autoencoder

methods. We briefly outline objective and subjective fusion indicators and use these evaluation indicators to test and evaluate several typical fusion methods. From the perspective of FLGC-fusion GAN, DDcGAN, and the latest technologies mentioned in this paper, DL has gradually developed and matured in the field of image fusion. However, in deep learning, which is widely used in the fusion of infrared and visible images, we still need to pay attention to the fusion effect and calculation choices. Furthermore, there will be more fusion technologies based on different methods, and deep learning methods will still be the popular trend in this application field.

**Author Contributions:** C.S. wrote the draft; C.Z. gave professional guidance and edited; N.X. gave advice and edited. All authors have read and agreed to the published version of the manuscript.

**Funding:** This research was funded by Major Technical Innovation Projects of Hubei Province (id: 2018ABA099); Innovation and Education Fund of the Science and Technology Development Center of the Ministry of Education of China (grant no. 2018A01038); National Science Fund for Youth of Hubei Province of China (grant no. 2018CFB408); Natural Science Foundation of Hubei Province of China (grant no. 2015CFA061); National Nature Science Foundation of China (grant no. 61272278).

**Conflicts of Interest:** The authors declare that they have no conflicts of interest.

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
