# Peer review of "Infrared and Visible Image Fusion Techniques Based on Deep Learning: A Review"

_electronics, doi:10.3390/electronics9122162_

Round 1

Reviewer 1 Report

This is a survey paper analyzing the existing infrared and visible image fusion methods based on deep learning. To improve the paper's quality, the following issues should be addressed carefully.

  • In the abstract, the authors write ”this paper proposes an advanced fusion method based on generative confrontation network”. However, I do not see any contents to describe the proposed method.
  • Some recent papers related to infrared and visible image fusion methods are missing. Please review the following recent papers.
    • (1). Yunfan Chen and Hyunchul Shin. "Multispectral image fusion based pedestrian detection using a multilayer fused deconvolutional single-shot detector." JOSA A 37, no. 5 (2020): 768-779.
    • (2). Yunfan Chen, Han Xie, and Hyunchul Shin. "Multi-layer fusion techniques using a CNN for multispectral pedestrian detection." IET Computer Vision 12, no. 8 (2018): 1179-1187.
    • (3) Jingchun Piao, Yunfan Chen, and Hyunchul Shin. "A new deep learning based multi-spectral image fusion method." Entropy 21, no. 6 (2019): 570.
    • (4) Ruichao Hou, Dongming Zhou, Rencan Nie, Dong Liu, Lei Xiong, Yanbu Guo, and Chuanbo Yu. "VIF-Net: an unsupervised framework for infrared and visible image fusion." IEEE Transactions on Computational Imaging 6 (2020): 640-651.
    • (5) Jiangtao Xu, Xingping Shi, Shuzhen Qin, Kaige Lu, Han Wang, and Jianguo Ma. "LBP-BEGAN: A generative adversarial network architecture for infrared and visible image fusion." Infrared Physics & Technology 104 (2020): 103144.
    • (6) Qilei Li, Lu Lu, Zhen Li, Wei Wu, Zheng Liu, Gwanggil Jeon, and Xiaomin Yang. "Coupled GAN with relativistic discriminators for infrared and visible images fusion." IEEE Sensors Journal (2019).
  • For a survey paper, the analysis of existing studies is poor. Improve the quality of explanation and analysis of existing methods in Section 2.
  • Some subjective evaluation metrics are missing in Section 3.1, such as Chen-Varshney Metric, Chen-Blum Metric, and Visual Information Fidelity.
  • It is necessary to introduce the existing available infrared and visible image fusion datasets, such as INO’s Video Analytics Dataset and OSU Color-Thermal Database.
  • The quality of analysis for experimental results should be improved. For example, in Table 5, some existing state-of-the-art methods have not been compared.

Author Response

Response to Reviewer 1 Comments

Dear reviewer:

We would like to thank Computer Science & Engineering for giving us the opportunity to revise our manuscript. All the comments are valuable and helpful for revising and improving our paper and the important guiding significance to our research. We have revised our manuscript based on your comments. The main correction in the paper and the respond to the reviewer's comments are as following:

Point 1: In the abstract, the authors write, “this paper proposes an advanced fusion method based on generative confrontation network”. However, I do not see any contents to describe the proposed method.

Response 1: Thank you very much for your comment. According to your Suggestions, we have reconsidered the starting point of this paper. This paper reviews image fusion methods based on deep learning and the paper will summarize the new methods proposed at present. We have revised the summary for the inexplicable parts, as shown in lines 10 to 17 on page 1. The revised contents are as follows:

“Infrared and visible image fusion technologies make full use of different image features obtained by different sensors, retain complementary information of the source images during the fusion process, and use redundant information to improve the credibility of the fusion image. In recent years, many researchers have used deep learning methods (DL) to explore the field of image fusion and found that applying DL has improved the time-consuming efficiency of the model and the fusion effect. However, DL includes many branches, and there is currently no detailed investigation of deep learning methods in image fusion. In this work, this survey reports on the development of image fusion algorithms based on deep learning in recent years.”

Point 2: Some recent papers related to infrared and visible image fusion methods are missing. Please review the following recent papers.

  • (1). Yunfan Chen and Hyunchul Shin. "Multispectral image fusion based pedestrian detection using a multilayer fused deconvolutional single-shot detector." JOSA A 37, no. 5 (2020): 768-779.
  • (2). Yunfan Chen, Han Xie, and Hyunchul Shin. "Multi-layer fusion techniques using a CNN for multispectral pedestrian detection." IET Computer Vision 12, no. 8 (2018): 1179-1187.
  • (3) Jingchun Piao, Yunfan Chen, and Hyunchul Shin. "A new deep learning based multi-spectral image fusion method." Entropy 21, no. 6 (2019): 570.
  • (4) Ruichao Hou, Dongming Zhou, Rencan Nie, Dong Liu, Lei Xiong, Yanbu Guo, and Chuanbo Yu. "VIF-Net: an unsupervised framework for infrared and visible image fusion." IEEE Transactions on Computational Imaging 6 (2020): 640-651.
  • (5) Jiangtao Xu, Xingping Shi, Shuzhen Qin, Kaige Lu, Han Wang, and Jianguo Ma. "LBP-BEGAN: A generative adversarial network architecture for infrared and visible image fusion." Infrared Physics & Technology 104 (2020): 103144.
  • (6) Qilei Li, Lu Lu, Zhen Li, Wei Wu, Zheng Liu, Gwanggil Jeon, and Xiaomin Yang. "Coupled GAN with relativistic discriminators for infrared and visible images fusion." IEEE Sensors Journal (2019).

Response 2: Thank you very much for your comment. We are very glad to receive the latest papers from you, which are very useful to us. The paper (1) can not be browsed in various academic indexes, only the abstract part can be seen, So I quoted it to line 74 of page 2 based on the abstract details. I have read and summarized the papers from (2) to (6). Now I have added the paper to the corresponding methods for a summary and analyzed it in each section's table. The revised contents are as follows:

(1): “In view of the above problems, the image fusion method based on deep learning can assign weights to the model through an adaptive mechanism [127].”

(2) Lines 168 to 178 on page 6, “In [109], Chen et al. used deep learning methods to fuse visible information and thermal radiation information in multispectral images. This method uses the Multi-Layer Fusion (MLF) area network in the image fusion stage. In this way, pedestrians can be detected at different ratios under unfavorable lighting (such as shadows, overexposure, or night) conditions. To be able to handle targets of various sizes, prevent the omission of some obscure pedestrian information. In the region extraction stage, MLF-CNN designed a multiscale Region Proposal Network (RPN) [110] to fuse infrared and visible light information and use summation fusion to fuse two convolutional layers.”

(3) Lines 212 to 218 on page 7, “In [113], Piao et al. designed an adaptive learning model based on the Siamese network, which automatically generates the corresponding weight map through the saliency of each pixel in the source image to reduce the number of traditional fusion rules. The parameter redundancy problem. This paper uses a three-level wavelet transform to decompose the source image into a low-frequency weight map and a high-frequency weight map. The scaled weight map is used to reconstruct the wavelet image to obtain the corresponding fused image. This result is more consistent with the human visual perception system. There are fewer undesirable artifacts.”

(4) Lines 175 to 178 on page 6, “In [111], to solve the lack of label dataset, Hou et al. used a mixed loss function. The thermal infrared image and the visible image were adaptively merged by redesigning the loss function, and noise interference was suppressed. This method can retain salient features and texture details with no apparent artifacts and have high computational efficiency.”

(5) Lines 250 to 254 on page 9, “In [114], Xu et al., based on Local Binary Pattern (LBP) [115], intuitively reflected the edge information of the image by comparing the values between the central pixel and the surrounding eight pixels to generate a fusion image with richer boundary information. The discriminator encodes and decodes the fused image and each source image, respectively, and measures the difference between the distributions after decoding.”

(6) Lines 254 to 257 on page 9, “In [116], Li et al. used the pre-fused image as the label strategy so that the generator takes the pre-fused image as the benchmark in the generation process so that the image fused by the generator can effectively and permanently retain the rich texture in the visible image and the thermal radiation information in the infrared image.”

Point 3: For a survey paper, the analysis of existing studies is poor. Improve the quality of explanation and analysis of existing methods in Section 2.

Response 3: Thank you very much for your comment. Based on your suggestions, we have reanalyzed the existing methods cited in Section 2. Also, the main methods involved in Section 2 are listed in Table 1 so that the general idea of all methods can be easily understood. In each section of 2.1, 2.2, 2.3, and 2.4, we cite the latest papers you have provided, and we also collect some of the latest articles to summarize.

Point 4: Some subjective evaluation metrics are missing in Section 3.1, such as Chen-Varshney Metric, Chen-Blum Metric, and Visual Information Fidelity.

Response 4: Thank you very much for your comment. After receiving your suggestions, we consulted professional materials in recent years. In previous articles related to image fusion, the three indexes of Chen-Varshney Metric, Chen-Blum Metric, and Visual Information Fidelity were listed as objective evaluation indexes. Therefore, we added these three methods in Table 5 of Section 3.2 and introduced them in sections 12) and 13) on page 14, respectively. Besides, in Section 3.1, lines 311 to 312 on page 10, we list two subjective evaluation approaches. The modified content is as follows:

Section 3.1: “The subjective evaluation can be further divided into the absolute evaluation and relative evaluation, which is implemented by a widely-recognized five-level quality scale and obstacle scale [128].”

Point 5: It is necessary to introduce the existing available infrared and visible image fusion datasets, such as INO’s Video Analytics Dataset and OSU Color-Thermal Database.

Response 5: Thank you very much for your comment. In section 4 of page 14, we have added the introduction of datasets about infrared and visible image fusion, including the INO’s Video Analytics Dataset and OSU Color-Thermal Database you mentioned. The added content is:

Lines 434 to 439 on page 14, “At present, there are several existing visible and infrared image fusion data sets, including OSU color thermal database [121], TNO image fusion Dataset [122], INO Video Analytics Dataset, VLIRVDIF [123], VIFB [124], and RGB-NIR Scene Dataset [125]. Table 5 summarizes the preliminary information about these datasets. In fact, except for OSU, there are not many image pairs in TNO and VLIRVDIF. However, the lack of a code library, evaluation metrics, and the results of these datasets makes it difficult to evaluate the latest technologies against them.”

Table 6. Details of existing infrared and visible image fusion datasets.

Name

Image/Video pairs

Image type

Resolution

Year

TNO

63 image pairs

Multispectral

Various

2014

VLRVDIF

24 video pairs

RGB, Infrared

720´480

2019

OSU Color-Thermal Database

6 video pairs

RGB, Infrared

320´240

2005

VIFB

21 image pairs

RGB, Infrared

Various

2020

INO-Video Analytics Dataset

223 image pairs

RGB, Infrared

Various

2012

RGB-NIR Scene Dataset

477 image pairs

RGB, Near-infrared (NIR)

1024´768

2011

Point 6: The quality of analysis for experimental results should be improved. For example, in Table 5, some existing state-of-the-art methods have not been compared.

Response 6: Thank you very much for your comment. Your suggestion has always been on my mind. Currently, there are many excellent articles. We found the code library of DDcGAN, carried out experiments, and added the experimental results to Figure 8, Table 7, and Table 8. However, there is a lack of a code library that can gauge the state-of-the-art. We cannot obtain real experimental results through experiments for effective comparison. If the relevant code base becomes rich in the future, we will try to remedy the regret.

Reviewer 2 Report

This paper presents a survey on infrared and visible image fusion methods based on deep learning approach. Although this is the good attempt to summarize different state of the art techniques, but the major structure of the paper is not appropriate for the understanding. I would recommend following improvements:

  1. I couldn’t find any context of deep learning used in fusion of infrared and visible images in the abstract. Pleaser rewrite the abstract and summarize the importance and advantages of deep learning over existing methods.
  2. Line 10, sentence is not clear.
  3. Line 15, model accuracy and speed of what?
  4. Line 17, The abstract claims that the new method is proposed but I couldn’t find any detail in the text.
  5. In introduction, Line 51 onwards, paragraph are not clear and linked with previous text. Suddenly one sentence not related with the previous context appears.
  6. Line 63, deep learning with low computational cost. What do you mean? Please explain.
  7. Line 93, please explain artificially designed extraction methods.
  8. Line 98 launch is not an appropriate word.
  9. Section 2.1, Add tables for the different methods and divide section in different sub sections to summarize different techniques.
  10. Section 3.2,The detail explanation of evaluation metrics does not contribute to the scope of this paper. The main scope is to evaluate deep learning based image fusion methods, I would suggest to add the technical background of some of these methods.
  11. Section 4, I couldn’t find explanation of the proposed method which is evaluated in section 4.2 onwards.

Author Response

Response to Reviewer 2 Comments

Dear reviewer:

We would like to thank Computer Science & Engineering for allowing us to revise our manuscript. All the comments are valuable and helpful for revising and improving our paper and the important guiding significance of our research. We have revised our manuscript based on your comments. The main correction in the paper and the response to the reviewer's comments are as following:

Point 1: I couldn’t find any context of deep learning used in fusion of infrared and visible images in the abstract. Pleaser rewrite the abstract and summarize the importance and advantages of deep learning over existing methods.

Response 1: Thank you very much for your comment. Based on your suggestions, we carefully considered the starting point of this article, rewritten the summary, and summarized the importance of deep learning for image fusion. The revised contents are as follows:

Lines 10 to 23 on page 1 of 25, “Infrared and visible image fusion technologies make full use of different image features obtained by different sensors, retain complementary information of the source images during the fusion process, and use redundant information to improve the credibility of the fusion image. In recent years, many researchers have used deep learning methods (DL) to explore the field of image fusion and found that applying DL has improved the time-consuming efficiency of the model and the fusion effect. However, DL includes many branches, and there is currently no detailed investigation of deep learning methods in image fusion. In this work, this survey reports on the development of image fusion algorithms based on deep learning in recent years. Specifically, this paper first conducts a detailed investigation on the fusion method of infrared and visible images based on deep learning, compares the existing fusion algorithms qualitatively and quantitatively with the existing fusion quality indicators, and discusses various fusions. The main contribution, advantages, and disadvantages of the algorithm. Finally, the research status of infrared and visible image fusion is summarized, and future work has prospected. This research can help us realize many image fusion methods in recent years and lay the foundation for future research work.”

Point 2: Line 10, sentence is not clear.

Response 2: Thank you very much for your comment. The original line 10 belongs to the Abstract section, but the summary has been modified. See Response 1 for details.

Point 3: Line 15, model accuracy and speed of what?

Response 3: Thank you very much for your comment. The accuracy of infrared and visible image fusion direction is mainly based on the subjective and objective evaluation indexes in Section 3.1 and 3.2. Due to the lack of the latest lack of code Library that can gauge the state-of-the-art, we cannot obtain real experimental results through experiments for effective comparison. Therefore, we conducted experiments on some models. The accuracy of image fusion in various aspects is shown in Table 7, and the speed is shown in Table 8.

Table 7. The average evaluation index value with different methods.

Methods

SF

EN

CC

COSIN

SD

SSIM

QAB/F

Deep Fuse

6.9603

6.7448

0.4182

0.9706

44.0915

0.4259

0.4396

CVT

6.9696

6.2151

0.4023

0.9118

28.7006

0.394

0.5065

DTCWT

6.9266

6.6617

0.3892

0.9549

32.6006

0.4652

0.5359

GFF

7.0902

7.1238

0.3711

0.8683

43.2243

0.3763

0.6168

GTF

7.2161

7.1464

0.3622

0.9739

44.3197

0.3858

0.4154

Fusion GAN

7.4707

7.377

0.4473

0.9715

48.5708

0.4132

0.6237

FLGC-Fusion GAN

8.0068

7.5199

0.4861

0.9801

50.6215

0.4541

0.6941

DDcGAN

7.9142

7.4589

0.5558

0.9847

50.6318

0.5093

0.6434

Table 8. Run time comparison of six methods on the TNO datasets. The Fusion GAN, FLGC-Fusion GAN, and DDcGAN are performed on GPU while others are performed on the CPU. Each value denotes the mean of run time of a specific method on a dataset (unit: second).

Method

TNO

CVT

1.46

DTCWT

3.32´10-1

GFF

3.18´10-1

Deep Fuse

4.27´10-2

GTF

4.82

Fusion GAN

4.61´10-2

FLGC-Fusion GAN

3.07´10-2

DDcGAN

5.19´10-2

Point 4: Line 17, The abstract claims that the new method is proposed but I couldn’t find any detail in the text.

Response 4: Thank you very much for your comment. Since this article is presented in the form of a review, it should summarize the latest methods in this field. In conjunction with your comments, we have repositioned the starting point of this review. See Response 1 for details.

Point 5: In introduction, Line 51 onwards, paragraph are not clear and linked with previous text. Suddenly one sentence not related with the previous context appears.

Response 5: Thank you very much for your comment. Based on your comments, we have reorganized the thinking of this paragraph. This paragraph is based on the two major issues to be considered in image fusion mentioned above, which mainly involve fusion algorithms. Therefore, this sentence introduces fusion algorithms in detail and the three challenges involved in each type of fusion algorithm, briefly introduced in this paragraph. The revised contents are as follows:

Lines 48 to 56 on page 2 of 25, “Current fusion algorithms can be divided into seven categories, namely, multi-scale transform [17], sparse representation [18], neural network [19], subspace [20], saliency [21], hybrid models [22], and deep learning [23]. Each type of fusion method involves three key challenges, i.e., image transform, activity-level measurement, and fusion rule designing [24]. Image transformation includes different multiscale decomposition, various sparse representation methods, non-downsampling methods, and a combination of different transformations. The goal of activity level measurement is to obtain quantitative information to assign weights from different sources [3]. The fusion rules include the big rule and the weighted average rule, the essence of which plays the role of weight distribution [23].”

Point 6: Line 63, deep learning with low computational cost. What do you mean? Please explain.

Response 6: Thank you very much for your comment. Thank you for your suggestion. The sentence was originally explained too briefly, but the reason has now been added. The added contents are as follows:

Lines 69 to 75 on page 2 of 25, “According to the different imaging characteristics of the source image, the selection of traditional manually designed fusion rules to represent the fused image in the same way will lead to the lack of diversity of extracted features, which may bring artifacts to the fused image. Moreover, for multi-source image fusion, manual fusion rules will make the method more and more complex. In view of the above problems, the image fusion method based on deep learning can assign weights to the model through an adaptive mechanism [127]. Compared with the design rules of traditional methods, this method greatly reduces the calculation cost, which is crucial in many fusion rules.”

Point 7: Line 93, please explain artificially designed extraction methods.

Response 7: Thank you very much for your comment. We had already explained the artificially designed extraction method in detail when we modified Point 6, and we also modified some of the content in Section 2. The revised contents are as follows:

Lines 98 to 105 on page 3 of 25, “However, these artificially designed extraction methods make the image fusion problem more complicated due to their limitations. In order to overcome the limitations of traditional fusion methods, deep learning methods are introduced for feature extraction. In recent years, with the development of deep learning, several fusion methods based on CNN, GAN, Siamese network, and autoencoder have appeared in the field of image fusion. The main fusion methods involved in this section are listed in Table 1 by category. Image fusion results based on deep learning have good performance, but many methods also have apparent challenges. Therefore, we will introduce the details of each method in detail.”

Point 8: Line 98 launch is not an appropriate word.

Response 8: Thank you very much for your comment. I'm very sorry for the trouble, we have corrected it in time. Now the sentence in which the word was changed has been revised. The revised contents are as follows:

Lines 104 to 105 on page 3 of 25, “Therefore, we will introduce the details of each method in detail.”

Point 9: Section 2.1, Add tables for the different methods and divide section in different sub sections to summarize different techniques.

Response 9: Thank you very much for your comment. According to your suggestions, we designed Table 1 to sort out most of the reference methods in Section 2 of the article. In this way, you can first clearly understand the general direction and use the strategy of each method. The added contents are as follows:

Lines 102 to 103 on page 3 of 25, “The main fusion methods involved in this section are listed in Table 1 by category.”

Table 1. The Overview of some DL-based fusion methods.

Families of fusion methods

Ref.

Innovation

CNN mehod of DL

[30]

Dense Net.

[29]

VGG-19, L1 norm, weighted average strategy, Maximum selection strategy.

[32]

ZCA-Zero-phase component analysis, L1-norm, Weighted average strategy.

[31]

Minimize the total change.

[34]

Lastic weight consolidation.

[35]

Perceptual loss, Use two convolutional layers to extract image features, Weight sharing strategy.

[36]

Adaptive information preservation strategy.

[109]

MLF-CNN, Weighting summation strategy.

[111]

Mixed loss function (M-SSIM loss,TV loss), Adaptive VIF-Net.

Siamese network of DL

[39]

Pixel-level image fusion, Feature tracking.

[38]

Fusion strategy of local similarity, Weighted average.

[112]

Dual siamese network, Weight sharing strategy.

[113]

saliency map, Three-level wavelet transform.

GAN of DL

[24]

The confrontation between generator and discriminator.

[41]

Adversarial generation network with dual discriminators.

[42]

Detail loss, Target edge loss.

[40]

Learnable group convolution.

[114]

local binary pattern.

[116]

Pre-fused image as the label.

Autoencoder of DL

[45]

Combination of autoencoder and Dense network.

[41]

Automatic coding feature extraction strategy of generator.

[130]

RGB encoder, infrared encoder, decoder used to restore the resolution of the feature map.

Point 10: Section 3.2, The detail explanation of evaluation metrics does not contribute to the scope of this paper. The main scope is to evaluate deep learning based image fusion methods, I would suggest to add the technical background of some of these methods.

Response 10: Thank you very much for your comment. According to your suggestion, I have consulted the relevant papers. This point is understood as follows: the performance of image fusion techniques applied in different fields will vary greatly due to different fusion methods. Different fusion techniques have different emphases, such as the targeted preference for contrast, detailed information, and removal of artifacts. Therefore, in each evaluation index introduced in Section 3.2, the emphasis of different fusion index evaluation, and the source of each method are described in detail. Image fusion and fusion index are inseparable.

Point 11: Section 4, I couldn’t find explanation of the proposed method which is evaluated in section 4.2 onwards.

Response 11: Thank you very much for your comment. We quote the source of each method in the qualitative evaluation notes of Figure 8 in Section 4.2. The revised contents are as follows:

Lines 484 to 485 on page 17 of 25, “(a) the visible image, (b) the infrared image, (c) Deep Fuse [30], (d) CVT [75], (e) DTCWT [129], (f) GFF [17], (g) GTF [92], (h) Fusion GAN [24], (i) FLGC-Fusion GAN [40], (j) DDcGAN [41].”

Reviewer 3 Report

Dear Authors;

I have read your review paper which investigated and intensely studied the infrared and visible image fusion methods based on deep learning and their applications in recent years. I send you some questions and comments and I wish you give the answers and make the required modifications:

1- Page 7 of 20, Line 233, rewrite this sentence (this method has regularization it can reduce the over fitting of the model), why It not it, and clarify which model you mean?
2- Page 10 of 20, You have to define what are X and Y for equation 6.
3- Page 10 of 20, Line 315, give the reference of this sentence (A high-contrast fusion image will produce a larger SD, which means ...).
4- Page 12 of 20, Line 397, rewrite this sentence (After sub-sampling the chrominance component to reduce the chrominance component), and please give the reference of this sentence (the naked eye will not perceive image quality change)

Author Response

Response to Reviewer 3 Comments

Dear reviewer:

We would like to thank Computer Science & Engineering for giving us the opportunity to revise our manuscript. All the comments are valuable and helpful for revising and improving our paper and the important guiding significance of our research. We have revised our manuscript based on your comments. The main correction in the paper and the respond to the reviewer's comments are as following:

Point 1: Page 7 of 20, Line 233, rewrite this sentence (this method has regularization it can reduce the over fitting of the model), why It not it, and clarify which model you mean?

Response 1: Thank you very much for your comment. We carefully revised this sentence, clearly introduced the meaning of this paragraph, and pointed out its source. The revised contents are as follows:

Lines 273 to 274 on page 9 of 25, “(3) this dense connection method has a regularization effect, which can reduce overfitting caused by too many parameters [44].”

Point 2: Page 10 of 20, You have to define what are X and Y for equation 6.

Response 2: Thank you very much for your comment. In

www.latexlive.com/#QyUyMEMoWCwlMjBZKT0lNUNmcmFjJTdCJTVDb3BlcmF0b3JuYW1lJTdCQ292JTdEKFgsJTIwWSklN0QlN0IlNUNzcXJ0JTdCJTVDb3BlcmF0b3JuYW1lJTdCVmFyJTdEKFgpJTIwJTVDb3BlcmF0b3JuYW1lJTdCVmFyJTdEKFkpJTdEJTdE,

X represents the infrared or visible image, and Y represents the fused image. However, equation 6 is a little simplified to understand, so I found a new formula describing the Correlation coefficient (CC) to replace it. The revised contents are as follows:

Lines 352 to 364 on page 12 of 25, “The CC measures the degree of linear correlation between a fused image and infrared and visible images and is mathematically defined as follows:

www.latexlive.com/#QyUyMEM9JTVDZnJhYyU3QiU1Q2xlZnQocl8lN0JJLCUyMEYlN0Qrcl8lN0JWLCUyMEYlN0QlNUNyaWdodCklN0QlN0IyJTdE

(6)

www.latexlive.com/#cl8lN0JYLCUyMEYlN0Q9JTVDZnJhYyU3QiU1Q3N1bV8lN0JpPTElN0QlNUUlN0JIJTdEJTIwJTVDc3VtXyU3Qmo9MSU3RCU1RSU3QlclN0QoWChpLCUyMGopLSU1Q2JhciU3QlglN0QpKEYoaSwlMjBqKS0lNUNiYXIlN0JGJTdEKSU3RCU3QiU1Q3NxcnQlN0IlNUNzdW1fJTdCaT0xJTdEJTVFJTdCSCU3RCUyMCU1Q3N1bV8lN0JqPTElN0QlNUUlN0JXJTdEKFgoaSwlMjBqKS0lNUNiYXIlN0JYJTdEKSU1RSU3QjIlN0QlNUNsZWZ0KCU1Q3N1bV8lN0JpPTElN0QlNUUlN0JIJTdEJTIwJTVDc3VtXyU3Qmo9MSU3RCU1RSU3QlclN0QoRihpLCUyMGopLSU1Q2JhciU3QkYlN0QpJTVFJTdCMiU3RCU1Q3JpZ2h0KSU3RCU3RA==

(7)

where X represents IR or VIS image.`X and `F denotes the source image and fused image F average pixel values, H and W stand for the length and width of the test image. The larger fusion image of CC is closely related to source images and the better the fusion performance.”

Point 3: Page 10 of 20, Line 315, give the reference of this sentence (A high-contrast fusion image will produce a larger SD, which means ...).

Response 3: Thank you very much for your comment. According to your suggestion, we have added the quotation of the sentence. The revised contents are as follows:

Lines 363 to 364 on page 12 of 25, “A high-contrast fusion image will produce a larger SD, which means that the fusion image has a clear contrast [57].”

Point 4: Page 12 of 20, Line 397, rewrite this sentence (After sub-sampling the chrominance component to reduce the chrominance component), and please give the reference of this sentence (the naked eye will not perceive image quality change)

Response 4: Thank you very much for your comment. We have modified this sentence and provided a reference for it. The revised contents are as follows:

Lines 466 to 468 on page 15 of 25, “The human eye is more sensitive to the Y component of the image, so after sub-sampling the chrominance component Cb or Cr to reduce the chrominance component, the naked eye will not perceive a significant change in image quality [126].”

Round 2

Reviewer 2 Report

Most of the highlighted points are addressed in the revise version. However I think conclusion needs improvement. The starting sentence is not necessary. You didn't mention your conclusion about different techniques which one is better or some specific points. Normally at the end of review you mention what is your opinion or findings about these methods. Line 559, which field? Please add some specific points about the results or findings after the comparison of different methods.

Author Response

Dear reviewer:

We would like to thank Computer Science & Engineering for allowing us to revise our manuscript. All the comments are valuable and helpful for revising and improving our paper and the important guiding significance of our research. We have revised our manuscript based on your comments. The main correction in the paper and the response to the reviewer's comments are as following:

Point 1: Most of the highlighted points are addressed in the revise version. However I think conclusion needs improvement. The starting sentence is not necessary. You didn't mention your conclusion about different techniques which one is better or some specific points. Normally at the end of review you mention what is your opinion or findings about these methods. Line 559, which field? Please add some specific points about the results or findings after the comparison of different methods.

Response 1: Thank you very much for your comments. According to your suggestion, we have improved the conclusion. Furthermore, we need to explain that because we want to reflect clear ideas in the article, our views and findings in this area are mainly reflected in Section 5. In addition, in Section 2 of this article, we summarize the methods presented in this article. Section 4.2 illustrates the advantages and disadvantages of several typical fusion models through experiments and summarizes them in the text. The revised contents are as follows:

Lines 546 to 559 on page 19 of 25, “The application of DL-based techniques to visible and infrared image fusion has been progressing at a fast rate in recent years. However, due to the complexity of application scenarios and the pursuit of computational efficiency and fusion effect, different applications of IR and VI image fusion still need to be further improved, and there are also potential development directions. This paper reviews the latest developments in DL-based image fusion technology and summarizes the issues that should be improved in this field in the future. This review investigates infrared and visible image fusion methods based on DL in recent years. These methods are mainly divided into four categories: CNN-based methods, GAN-based methods, Siamese Network-based methods, and Autoencoder methods. We briefly outline objective and subjective fusion indicators and use these evaluation indicators to test and evaluate several typical fusion methods. From the perspective of FLGC-Fusion GAN, DDcGAN, and the latest technologies mentioned in this paper, DL has gradually developed and matured in the field of image Fusion. But in deep learning, which is widely used in the fusion of infrared and visible images, we still need to pay attention to the fusion effect and calculation choices.”
